DATA RELEASE

# LT1, an ONT long-read-based assembly scaffolded with Hi-C data and polished with short reads

Hui-Su Kim[1,*], Asta Blazyte[1,2], Sungwon Jeon[1,2,3], Changhan Yoon[1,2], Yeonkyung Kim[1,3], Changjae Kim[3], Dan Bolser[4], Ji-Hye Ahn[3], Jeremy S. Edwards[5] and Jong Bhak[1,2,3,4,*]

1 Korean Genomics Center (KOGIC), Ulsan National Institute of Science and Technology (UNIST), Ulsan 44919, Republic of Korea
2 Department of Biomedical Engineering, College of Information and Biotechnology, Ulsan National Institute of Science and Technology (UNIST), Ulsan 44919, Republic of Korea
3 Clinomics LTD, Ulsan National Institute of Science and Technology (UNIST), Ulsan 44919, Republic of Korea
4 Geromics Ltd., Suite 1, Frohock House, 222 Mill Road, Cambridge, CB1 3NF, UK
5 Department of Chemistry and Chemical Biology, University of New Mexico, Albuquerque, NM 87131, USA

## ABSTRACT

We present LT1, the first high-quality human reference genome from the Baltic States. LT1 is a female *de novo* human reference genome assembly, constructed using 57× nanopore long reads and polished using 47× short paired-end reads. We utilized 72 GB of Hi-C chromosomal mapping data for scaffolding, to maximize assembly contiguity and accuracy. The contig assembly of LT1 was 2.73 Gbp in length, comprising 4490 contigs with an NG50 value of 12.0 Mbp. After scaffolding with Hi-C data and manual curation, the final assembly has an NG50 value of 137 Mbp and 4699 scaffolds. Assessment of gene prediction quality using Benchmarking Universal Single-Copy Orthologs (BUSCO) identified 89.3% of the single-copy orthologous genes included in the benchmark. Detailed characterization of LT1 suggests it has 73,744 predicted transcripts, 4.2 million autosomal SNPs, 974,616 short indels, and 12,079 large structural variants. These data may be used as a benchmark for further in-depth genomic analyses of Baltic populations.

**Submitted:** 07 December 2021

\* Corresponding authors. E-mail:
jongbhak@genomics.org;
hskim3824@gmail.com

Preprint submitted at https://doi.org/10.1101/2021.04.05.438426

**Subjects** Genetics and Genomics, Genetics, Personalized Medicine

## DATA DESCRIPTION

The Baltic States comprise three countries, Lithuania, Latvia, and Estonia, located on the eastern coast of the Baltic Sea, where northern, eastern, and central European regions converge. People of the Baltic States share a regional identity [1] and an endemic Landsteiner–Wiener blood group, a blood biomarker that is found in high frequencies only in these three populations [2]. The Baltic populations were shaped by multiple genetic influxes such as from Anatolia, Mesolithic Western hunter-gatherers, and central Europe [3, 4], as well as by a complex recent history resulting from wars and annexations. Despite the significance of the Baltic region, the genetic makeup of Baltic Sea region populations has not yet been extensively studied, especially compared with those of central or southern Europe [3].

Lithuanian and Latvian people have been consistently reported as genetically homogeneous [5–8], sharing a very similar genomic structure [8, 9]. Until now, genomic

research in Lithuanians has mainly utilized single nucleotide polymorphism (SNP) genotyping [5, 10–13] or exome sequencing [14, 15]. To expand the scope of analyses and increase the possibility of new findings, whole-genome sequencing (WGS) using long-read technologies is an optimal solution; it enables the discovery of novel genomic variations [16], reveals accurate breakpoints of the structural variations (SV), and covers some of the complex repeat regions [17–19]. Consequently, resolving haplotypes is also relevant to high-quality *de novo* whole-genome assembly and phasing [17].

## CONTEXT

Here, we utilized PromethION, the long-read sequencing platform from Oxford Nanopore Technologies (ONT), as a backbone to construct the first Lithuanian reference genome, LT1, using the genome of a healthy female with Lithuanian ancestry. The ONT PromethION long-read-based genome assembly was polished using BGI-500 short-reads and scaffolded by utilizing Hi-C chromatin conformation capture data.

The final assembly had an NG50 value of 138 megabase pairs (Mbp) and 4,699 scaffolds, covering 92.75% of GRCh38 (Genome Reference Consortium Human Build 38) [20]. SV analyses using long-read data identified 12,079 consensus SVs (confirmed by both SVIM [21] and NextSV2 [22]); however, more than half of the SVs (62.27% of 12,079) lacked information in all major databases, such as gnomAD [23], indicating that human SVs are an under-investigated area of population genetics.

Our high-quality assembly is the first step towards increasing the availability of human genome assemblies from the Baltic States and will serve as a valuable resource for further studies in population genomics.

## METHODS

### Sample preparation, library construction, and sequencing

A Lithuanian human (NCBI:txid9606) female with three generations of ethnic family history was recruited for sequencing. Standard ethical procedures were applied by the Genome Research Foundation with IRB-REC- 20101202 – 001. The volunteer signed an informed consent agreement, and a 20 mL blood sample was drawn using heparinized needles and collected into anticoagulant-containing tubes (dipotassium ethylenediaminetetraacetic acid; $K_2$ EDTA).

DNA was extracted from the single donor's peripheral blood (5 mL) using a DNeasy Blood & Tissue Kit from QIAGEN, according to the manufacturer's protocol. The quality and concentration of the extracted DNA were evaluated using NanoDrop™ One/OneC UV-Vis spectrophotometer (ThermoFisher Scientific™). Short-read whole genome sequencing and library construction were conducted by the Beijing Genomics Institute (BGI) on the BGISEQ-500 platform (RRID:SCR_017979) using DNBseq™ 100-basepair (bp) paired-end sequencing.

Sequencing libraries for long reads were prepared using the 1D ligation sequencing kit (SQK-LSK109) (Oxford Nanopore Technologies, UK) following the manufacturer's instructions. The products were quantified using the Bioanalyzer 2100 (Agilent, Santa Clara, CA, USA) and the raw signal data were generated on the PromethION R9.4.5 platform (Oxford Nanopore Technologies, UK). Base-calling from the raw signal data was carried out using a default ONT basecaller MinKNOW v19.05.1 with the Flip-Flop HAC (High Accuracy) model (Oxford Nanopore Technologies, UK).

**Table 1.** Statistics of long and short reads whole genome sequencing for LT1.

| Library type | Sequencing technology | Library name | Number of reads (n) | Total length of reads (bp) | NG50 (bp) |
|---|---|---|---|---|---|
| Short reads | BGISeq-500 | LT1_PE500 | 1,420,906,146 | 142,090,614,600 | 100 |
| Long reads | ONT PromethION | LT1_PT | 24,339,507 | 171,726,287,587 | 13,283 |
| Hi-C | Illumina NovaSeq | LT1_HiC | 935,185,202 | 140,277,780,300 | 150 |

### Hi-C sequencing data generation

Hi-C chromosome conformation capture data were generated using the Arima-HiC kit (A160105 v01, San Diego, CA, USA) with two restriction enzymes. To prepare LT1 samples for Hi-C analysis, white blood cells from the donated blood were harvested and cross-linked, as instructed by the manufacturer. One million cross-linked cells were used as input in the Hi-C protocol. Briefly, chromatin from cross-linked cells or nuclei was solubilized and then digested using multiple restriction enzymes that digest chromatin at ^GATC and G^ANTC. Digested ends were labeled using a biotinylated nucleotide and ligated. Ligation products were purified, fragmented, and size-selected using AMpure XP Beads. Biotinylated fragments were then enriched using Enrichment beads and Illumina-compatible sequencing libraries were constructed on end repair, dA-tailing, and adaptor ligation using a modified workflow of the Hyper Prep kit (KAPA Biosystems, Inc.). The bead-bound library was then amplified, and amplicons were purified using AMpure XP beads and subjected to deep sequencing. Sequencing of prepared Hi-C libraries was performed on the Illumina NovaSeq platform with read length of 150 bp by Novogene (Beijing, China).

### *De novo* assembly of the LT1 genome

To generate the *de novo* LT1 genome assembly, we prepared a bioinformatic pipeline including: a preprocessing step, contig assembly, map assembly, gene prediction, and post-analysis. The processes used in the pipeline are summarized in Figure 1.

A total of 142.09 gigabase pairs (Gbp) of short paired-end genomic raw reads were produced by the BGISEQ-500 sequencer, which resulted in a 47× sequencing depth of coverage (Table 1). Adapter sequences were trimmed from raw reads using Trimmomatic v0.36 (RRID:SCR_011848) [24] with parameters as 'ILLUMINACLIP:2:30:10 LEADING:3 TRAILING:3 SLIDINGWINDOW:4:20 HEADCROP:15 MINLEN:60'. Sequences from vectors and microbial contaminants were removed using BBtools suite v38.96 (RRID:SCR_016968) [25] and a merged database from Refseq databases (the prokaryotic Refseq and viral Refseq databases). For error correction, we used the tadpole.sh program of BBtools. After preprocessing, 106.29 Gbp cleaned reads were obtained.

In total, 172.22 Gbp raw long reads, giving 57× coverage, were produced from PromethION sequencing (Table 1). Ultra-long reads constituted 0.0075% of the long reads. Base-called raw reads with low quality were filtered by the default function of MinKNOW. Adapter sequences were trimmed using Porechop v.0.2.4 (RRID:SCR_016967) [26]. After preprocessing, 171.73 Gbp cleaned reads (57× coverage) was obtained.

One *de novo* assembly was performed using wtdbg2 v2.5 (RRID:SCR_017225) [27] with cleaned (filtered and trimmed) long reads. Parameters for the assembly were set as '−x ont −g 3g −L 5000'. For error correction of assembled contigs, we utilized a two-step strategy. First, correcting base-errors on contigs was carried out using four iterations of Racon v1.4.3 (RRID:SCR_017642) [28]. The parameters for Racon were '−m 8 −x −6 −g −8 −w 500'. As the



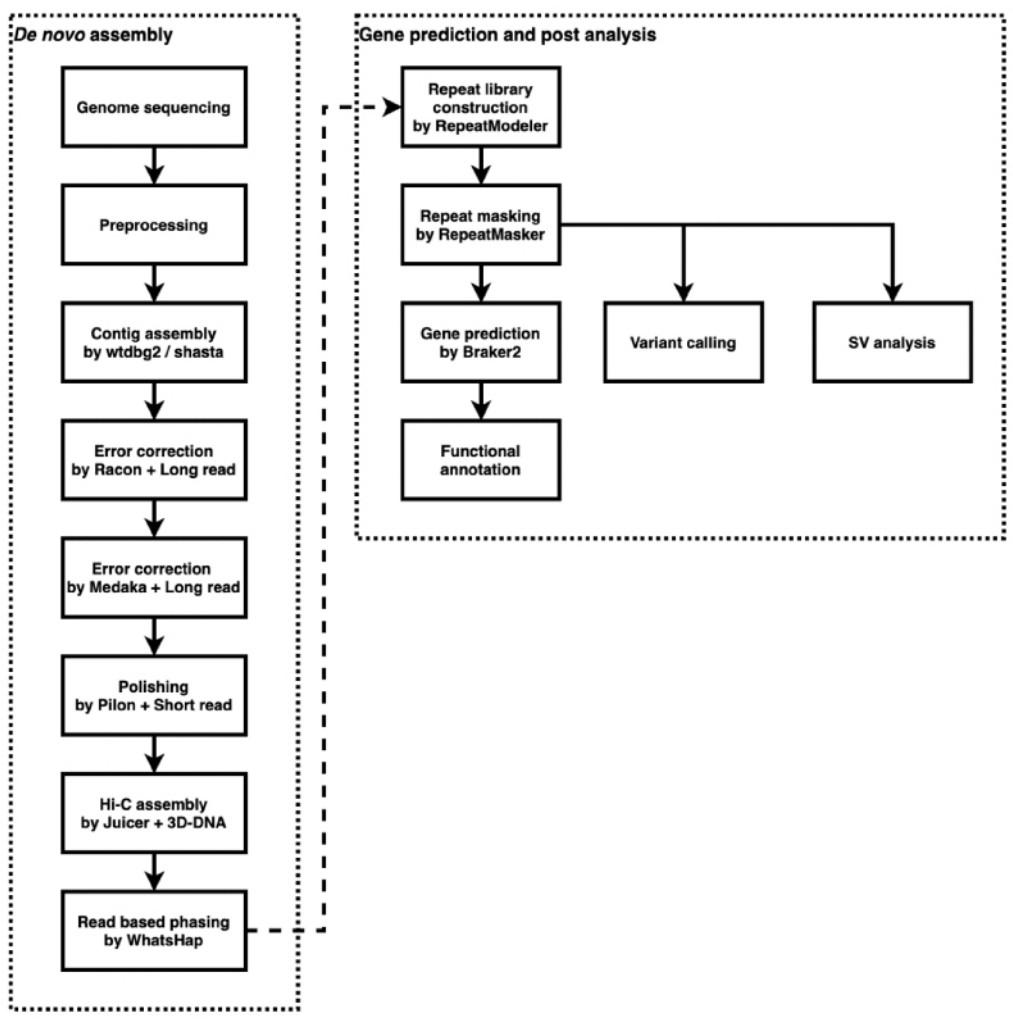

**Figure 1.** A bioinformatic pipeline for generating the LT1 genome assembly.

second step, error correction using Medaka v0.11.5 [29] was performed with a pre-trained model for Flip-Flop. To improve SNP and indel accuracy of the assembly, we polished the consensus with short reads using two rounds of Pilon v1.23 (RRID:SCR_014731) [30].

The second *de novo* assembly was performed using the Shasta v0.4.0 [31] assembler with the default parameters. For error correction of assembled contigs, MarginPolish v1.3 [32] and HELEN v0.0.1 [33] were used with default options. Using the two assemblers allowed us to compare the outcome of the two methods and choose one outperforming method for downstream analyses.

To generate a chromosome-level assembly for the LT1 genome, scaffolding with 72 GB of Hi-C reads was performed using Juicer (RRID:SCR_017226) [34] and 3D-DNA pipeline (RRID:SCR_017227) [35]. To correct misassemblies in the scaffolds, JBAT v1.11.08 (RRID:SCR_021172) [36] was used for manual curation. The map assembly was assessed using Nucmer v4.00beta2 (RRID:SCR_018171) [37] and Dot [38] against the human reference genome, GRCh38. All contigs from mitochondrial DNA were scaffolded to a complete mitochondrial (mtDNA) assembly using the same workflow as for the nuclear DNA.

Owing to the absence of LT1 parental genome data, a read-based phasing of the assembly was performed using Medaka and WhatsHap v1.0 [39] and shared on the LT1 webpage [40]. Since the variant calling module of Medaka includes variant calling and phasing steps with sequenced reads from ONT using WhatsHap, filtered and trimmed PromethION reads were mapped against the assembled scaffolds, and assembled scaffolds were phased using Medaka. As a result of read-based phasing, 2,299,025 variants were phased from 3,901,968 variants, and the number of phased blocks was 8879 (See Table S1 in GigaDB[41]). Phased genome sequences were extracted using Bcftools v1.9 (RRID:SCR_005227) [42] with a command-line of "bcftools consensus -H 1 -f reference.fasta phased.vcf.gz > haplotype1.fasta".

Merqury v1.0 [43] was used to assess LT1 assembly using *k*-mers. QUAST v5.0.2 [44] was used against GRCh38 and CHM13 v1.1 [45] to assess misassemblies on the LT1 assembly.

### Construction of repeat library and repeat masking
A repeat library was constructed for the LT1 genome. RepeatModeler v2.0.1 was used with LTRStruct (RRID:SCR_015027) [46]. Repeat masking was conducted using RepeatMasker v4.1.0 (RRID:SCR_012954) [47].

### Genome annotation
For annotation, protein-coding genes from GRCh38 were prepared. BRAKER v2.1.4 (RRID:SCR_018964) [48] with GeneMark-ES v4.38 (RRID:SCR_011930) [49] and Augustus v3.3.3 (RRID:SCR_008417) [50] were used to predict genes in LT1. Predicted genes were assessed using BUSCO v4.1.0 (RRID:SCR_015008) [51] with the mammalian orthologous gene set v10. Functional annotation of predicted genes was performed using NCBI-BLAST + v2.9.0 (RRID:SCR_004870) against the National Center for Biotechnology Information (NCBI) non-redundant protein database [52] and the Swiss-Prot database (RRID:SCR_002380) [53].

### Constructing a genome browser and BLAST database
To construct a genome browser, we first compiled all the data, including predicted gene models and evidence resources. The LT1 browser was built using JBrowse v1.16.9 (RRID:SCR_001004) [54]. A BLAST database for LT1 gene set v1 was built by SequenceServer v1.0.12 [55] and can be accessed via the Lithuanian genome webpage [40].

### Short indel and SNV calling
Variant calling was performed on preprocessed short reads using GATK v4.1.7 HaplotypeCaller (RRID:SCR_001876) [56] with –output-mode, EMIT_VARIANTS_ONLY and -stand-call-conf 30 settings. Preprocessed short reads were aligned to the GRCh38 reference genome using bwa aligner v0.7.15 (RRID:SCR_010910) [57] and sorting was carried out by SAMTOOLS v0.1.19 (RRID:SCR_002105) [58]. Duplicate marking and quality metric assessment were conducted using picard v1.3.2 (RRID:SCR_006525) [59]. Base quality scores from the alignment files were recalibrated using BaseRecalibrator and ApplyBQSR tools from GATK [60]. For SNV and indel recalibration, dbSNP v146 (RRID:SCR_002338) and Mills_and_1000G_gold_standard.indels sets were used, respectively.

### Structural variation analysis
SVs were identified from preprocessed nanopore reads using the Sniffles-based meta-pipeline NextSV2 [22] and the independent caller SVIM [21]. We used a NextSV2

**Table 2.** Contig assembly statistics.

|  | wtdbg2 assembly | Shasta assembly | wtdbg2 assembly (>1 kbp) | Shasta assembly (>1 kbp) |
|---|---|---|---|---|
| Number of contigs | 4,490 | 11,009 | 3,659 | 3,573 |
| Total length (bp) | 2,730,857,050 | 2,803,432,513 | 2,730,470,761 | 2,801,466,518 |
| NG50 (bp) | 12,010,256 | 6,252,833 | 13,411,207 | 6,731,385 |
| Maximum contig length (bp) | 65,254,771 | 43,332,451 | 65,254,771 | 43,332,451 |
| Gap (%) | 0.00 | 0.00 | 0.00 | 0.00 |
| GC content (%) | 40.82 | 40.88 | 40.82 | 40.89 |
| Quality value (QV) | 32.7665 | 28.6051 |  |  |

method, which employs Minimap2 (RRID:SCR_018550) [61], Sniffles v1.0.11 (RRID:SCR_017619) [62], and GRCh38 as the reference with all default settings. Default settings were also used for SVIM. Structural variants with genotypes 0/0, or supported by less than 10 reads, were filtered out and not presented in the final results. For SV merging and to estimate shared variants (as well as a union of the variants), we employed SURVIVOR v1.0.7 [63] with options 1000 (bp) for the window size and 30 (bp) for minimum SV length. SURVIVOR merging output was further used for AnnotSV v2.3 [64] multiple database annotation. Copy number variation (CNV) was estimated using CNVnator v.0.3.3 (RRID:SCR_010821) [65] with default parameters and output filtering settings $q_0 < 0.5$, where $q_0$ is the fraction of reads mapped with 0 (zero) mapping quality. The complete size distribution of the consensus deletions, insertions, and inversions was plotted in R using data.table, ggplot2 (RRID:SCR_014601) [66], and SiMRiv [67].

## DATA VALIDATION AND QUALITY CONTROL

### LT1 genome assembly statistics

The contig assembly using wtdbg2 resulted in 2.73 Gbp in 4,490 contigs with an NG50 of 12 Mbp. The Shasta assembly had 2.8 Gbp assembled into 11,009 contigs with an NG50 of 6.3 Mbp. Both contig assemblies were corrected using long-reads and polished with short-reads, as described in the Methods. The wtdbg2 assembly had higher contiguity and quality (the total number of contigs, NG50, quality value (QV) statistics); therefore, it was selected as the main assembly for the LT1 genome and subsequent analyses (Table 2).

Hi-C data were used for scaffolding (Figure 2). After scaffolding, we produced 4,700 scaffolds with a total length of 2.73 Gbp and an NG50 of 138 Mbp (Table 3), which includes an mtDNA. The number of scaffolds is higher than the original 4,490 contigs because we had to manually split some misassemblies that were found when we applied the Hi-C data. The longest scaffold was mapped to chromosome 2 and spanned 218 Mbp, which covers 92.6% of chromosome 2. To estimate the quality of the LT1 assembly, we compared it with GRCh38 and 'CHM13 Chromosome X v0.7' from T2T using Dot and QUAST. No significant misassemblies (such as translocations and inversions) were identified while analyzing NUCmer alignment plotted in Dot; notably, a comparison between LT1 and CHM13 chromosome X displayed a higher breadth of coverage (94.50%, Figure 3, supporting the fact that LT1 is a relatively high-quality assembly. The LT1 genome covered 92.75% of GRCh38 (excluding alternative contigs and chromosome Y in GRCh38), as shown in Figure 4 and Table 4. This shows that LT1 is significantly shorter than GRCh38 and there is still room for assembly improvement. QUAST analysis showed that the LT1 genome had, 479 and 842 misassemblies against GRCh38 and CHM13v1.1 (without ALT sequences), respectively (Table 4).

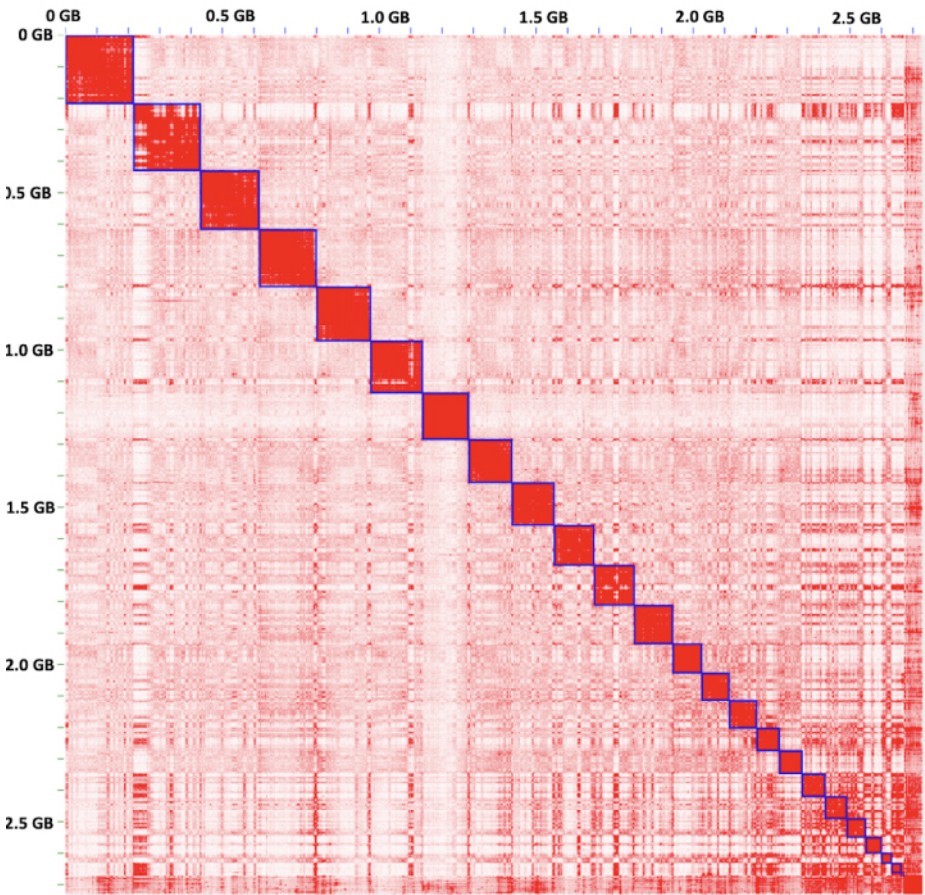

**Figure 2.** Contact map of Hi-C reads mapped against the LT1 assembly.

**Table 3.** Statistics of final LT1 assembly compared to GRCh38.

|  | LT1 genome v1 | GRCh38 p13 |
|---|---|---|
| Number of scaffolds | 4700 | 639 |
| Total length (bp) | 2,731,553,693 | 3,272,089,205 |
| NG50 (bp) | 136,648,960 | 145,138,636 |
| Maximum scaffold length (bp) | 218,797,018 | 248,956,422 |
| Gap (%) | 0.03 | 4.93 |
| GC content (%) | 40.84 | 41.04 |
| Quality value (QV) | 38.4098 |  |

The protein-coding gene set in LT1 comprised 73,744 transcripts when gene prediction was performed using BRAKER2. The total length of all the genes was 86.9 Mbp, with a median length of 1117 bp. This is much shorter than GRCh38. This could be because of the relatively lower base accuracy of the Nanopore Flip-Flop basecalling model. GC content in the coding region was 54.76% and the size of the longest gene, Titin, was 109,026 bp. To assess gene prediction and assembly quality, we performed a BUSCO analysis with the mammalian orthologous gene database v10. The ratio of complete single-copy orthologous genes was 89.3% (Table 5). Using BLAST, functional annotation of the LT1 genome against the non-redundant protein and Swiss-Prot database resulted in 47,015 (see Table S2 in

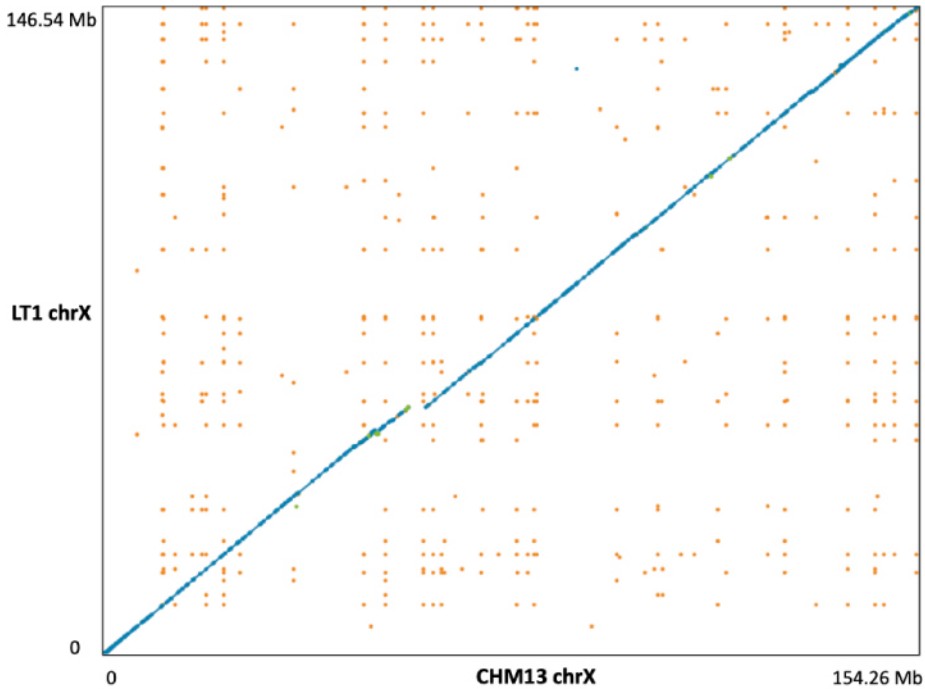

**Figure 3.** Comparison of chromosome X between LT1 and CHM13. Blue dots represent unique forward alignments and green dots unique reverse alignments, respectively. Orange dots denote repetitive alignments.

**Table 4.** The result of QUAST analysis, LT1 versus GRCh38 and CHM13 v1.1.

| Genome statistics | vs GRCh38 | vs CHM13v1.1 |
|---|---|---|
| Genome fraction (%) | 91.713 | 86.678 |
| Duplication ratio | 1.001 | 1 |
| Largest alignment | 33,029,577 | 42,496,445 |
| Misassemblies | 1479 | 842 |
| Relocations | 1139 | 789 |
| Translocations | 212 | 24 |
| Inversions | 128 | 29 |
| Local misassemblies | 606 | 366 |

GigaDB [41]) and 41,601 transcripts (see Table S3 in GigaDB [41]), respectively. BLAST analysis results are available on the LT1 genome webpage [40].

## Variant identification

LT1 is predicted to have 4,236,954 SNPs and 974,616 indels relative to GRCh38, based on mapping short reads against GRCh38. Private variants, here defined as variants unreported in dbSNPv.146, made up a significant portion of the identified variants (17.81%) (Table 6). Heterozygous deletions (30.72%) were the most underreported variant type.

For SVs, we use two SV callers, SVIM and NextSV2, and an SV analysis tool SURVIVOR. We identified a union of 31,167 SVs, of which 12,079 SVs were shared insertions, deletions, and inversions (Figure 5, Table 7).

The total number of deletions called by each tool after QC filtering differed minimally; however, the number of insertions identified by SVIM was two times higher than by

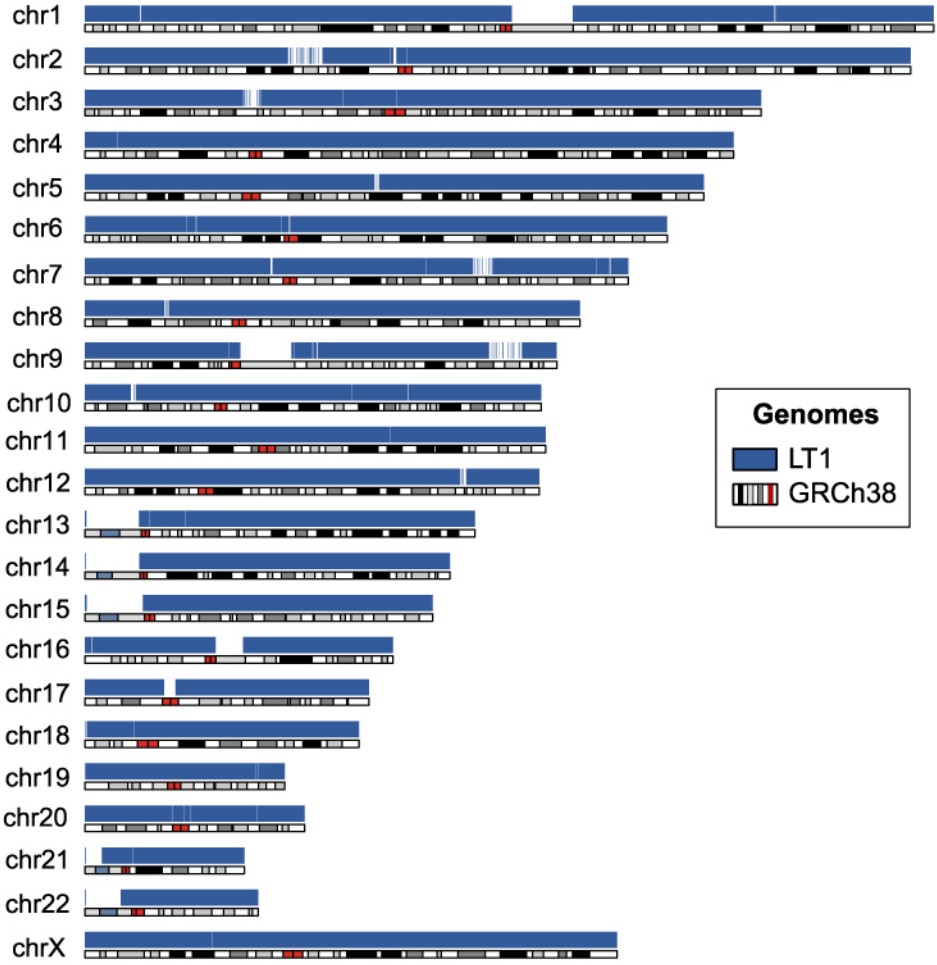

**Figure 4.** Alignment coverage of LT1 (blue) to the human reference genome GRCh38. The cytobands for the GRCh38 were pre-downloaded from UCSC by the karyoploteR package in R.

**Table 5.** LT1 and GRCh38 genome annotation.

|  | LT1 gene | GRCh38.p13 gene |
|---|---|---|
| Number of transcripts | 73,744 | 114,963 |
| Total length of transcripts (bp) | 86,273,106 | 228,562,356 |
| NG50 (bp) | 1887 | 2580 |
| Maximum transcript length (bp) | 109,026 | 107,973 |
| Gap (%) | 0.002 | 0.000 |
| GC content (%) | 54.76 | 51.02 |
| **BUSCO** | | |
| Complete (%) | 89.3 | 99.9 |
| Complete and single copy (%) | 49.2 | 27.5 |
| Complete and duplicated (%) | 40.1 | 72.4 |
| Fragmented (%) | 3.9 | 0.0 |
| Missing (%) | 6.8 | 0.1 |

NextSV2 (Table 7). This is probably because insertions, usually, are more difficult to call than deletions [68]. CNVnator detected 95 duplications, which was similar to the 97 detected

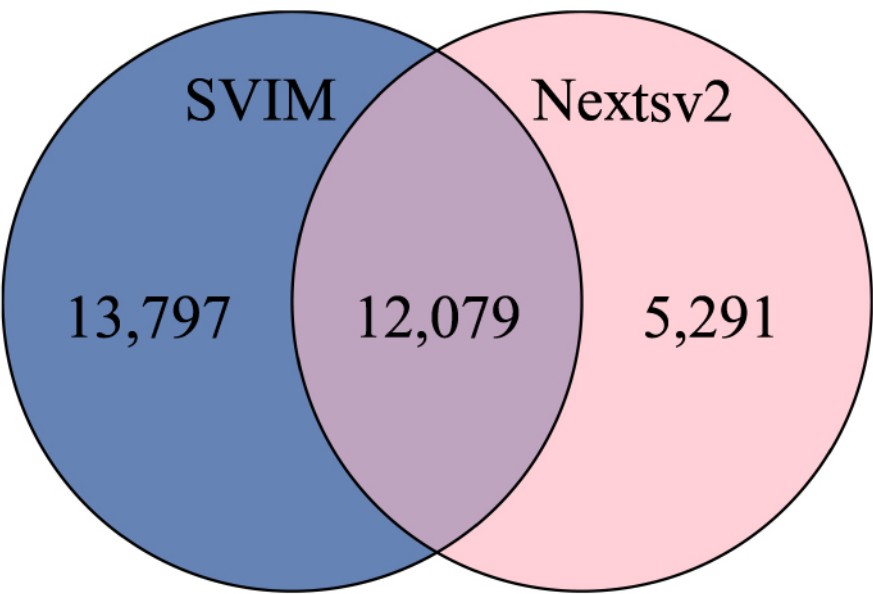

**Figure 5.** Consensus structural variations found in LT1.

**Table 6.** SNVs and short indels in LT1 reference genome based on GRCh38.

| | SNVs | Reported in dbSNP | % | Insertions | Reported in dbSNP | % | Deletions | Reported in dbSNP | % | Total variants | Reported in dbSNP | % |
|---|---|---|---|---|---|---|---|---|---|---|---|---|
| Total | 4,236,954 | 3,551,608 | 16.18 | 463,668 | 363,795 | 21.54 | 510,948 | 368,083 | 27.96 | 5,211,570 | 4,283,486 | 17.81 |
| Homozygous | 1,415,589 | 1,357,666 | 4.09 | 122,940 | 112,322 | 8.64 | 125,884 | 101,316 | 19.52 | | | |
| Heterozygous | 2,821,365 | 2,193,942 | 22.24 | 340,728 | 251,473 | 26.20 | 385,064 | 266,767 | 30.72 | | | |

by Nextsv2, when the minimum read support was 10; however, only one duplication was confirmed by both tools. Only a small fraction of the SVs could be attributed to any phenotype (10.19%) or found in the major databases such as gnomAD (RRID:SCR_014964) [23] (37.73%), despite the conservative 10 reads threshold to call an SV and confirmation by two different tools. Most SVs with phenotypes had a DGV (RRID:SCR_007000) [69] loss or gain of function annotation (58.21%), (Tables 7 and S6 in GigaDB[41]).

Almost 97% of SVs assigned to a genomic region were located in the introns [41]. Regardless of the SV type, small SVs (30–200 bp) constituted a significant fraction (64.77%), with another peak spiking around 300 bp (Figure 6). We confirmed that 81.92% of insertions forming this second peak (in the length range 250–350 bp) were ALU sequences. The longest insertion among the consensus SVs was detected on chromosome 7 in the LOC101928283 gene, spanning 1,003 bases. The largest deletion was in chromosome 1 (45,516 bases) and has been annotated in both the Database of Genomic Variants (DGV; RRID:SCR_007000) [68] and gnomAD (RRID:SCR_014964) [23] databases, despite lacking a precise annotation for a location or a clinical phenotype. It is predicted to be benign (Table S7 in GigaDB [41]).

## REUSE POTENTIAL

We present the first Lithuanian reference genome, LT1. ONT's PromethION long-read and BGI-500 short-read sequencing technologies were combined with Hi-C chromatin

**Table 7.** SV statistics based on GRCh38.

| | SVIM Duplications | NEXTSV2 Duplications | CNVnator Duplications | SVIM Inversions | NEXTSV2 Inversions | SVIM Deletions | NEXTSV2 Deletions | SVIM Insertions | NEXTSV2 Insertions | |
|---|---|---|---|---|---|---|---|---|---|---|
| Homozygous | *9* | 17 | NA | *3* | 26 | *2,686* | 2,313 | *2,822* | 1,392 | |
| Heterozygous | *15* | 80 | NA | *15* | 54 | *8,135* | 7,975 | *12,214* | 5,610 | |
| Total | *24* | 97 | 95 | *19* | 80 | *10,821* | 10,288 | *15,036* | 7,002 | |
| Survivor with merging parameter: merging max distance 1,000 | | | | | | | | | | Total |
| Overlap Homozygous | | 0 | | | *3* | | *482* | | *381* | 866 |
| Overlap Heterozygous | | 0 | | | *8* | | *3,845* | | *2,821* | 6,674 |
| Overlap Any (cases with disagreeing zigosity included) | | 0 | | | *18* | | *6,782* | | *5,279* | 12,079 |
| SVIM, NEXTSV2 SV intersection = 12,079 | | | | gnomAD annotations = 4,557 (37.73%) | Phenotype annotations **full** = 0, split = 1,231 (10.19%) | DGV annotations DGV loss 4,915 DGV gain 2,116* (58.21%) | dbVar annotations = 0 | (see sharedSV1000) | | |
| **Split' annotations (intersection)** | Intron | Exon | Total | % Intron | % Exon | Attributed to a gene | | | | |
| | 5,185 | 161 | 5,346 | 96.99 | 3.01 | 5,183 | | | | |

*Two columns were used for this database annotation. Bottom row denotes statistics which are not available with 'full' annotations and were presented based on 'split' type annotations (see Table S7 in GigaDB [41]).

conformation capture to complete the genome assembly. It was built with sufficient sequencing data to cover the genome and a high-quality assembly was constructed as the first reference genome from the Baltic States. BUSCO assessment revealed that LT1 gene prediction had more fragmented and missing genes against GRCh38 than initially expected.

Even though long DNA reads usually provide more accurate SV calling, our SV analyses with long-read data showed most SVs (62.27%) could not be annotated using currently available public databases. This indicates that SV is still an under-investigated area of population genetics. More ethnic references and regional genomic variation datasets (variomes) with phenotype association studies are needed to patch these remaining gaps in our knowledge to completely map and understand the biological features of the human genome structure.

This genome assembly could be used as a genomic reference representative of Lithuanian people in comparative genomics. Owing to its relatively high depth of coverage of long-read sequencing, this genome can be reused as a template to accurately map the autosomal and X chromosome genomic variation (both SNPs and SVs) of samples from the Baltic region. Moreover, here we provide the first long-read SV set for a healthy Lithuanian individual, which could be used in disease studies to filter out SVs in Lithuanian populations that do not cause any critical or early-onset disease phenotypes. Lastly, the unprocessed sequencing data; short reads, long reads, and Hi-C data, can be reused for any population genomics study.

## DATA AVAILABILITY

The whole genome sequence analyzed in this study has been deposited at the National Center for Biotechnology Information (NCBI) under BioProject ID PRJNA635750, in the NCBI BioSample database under accession number SAMN15052346, and in the NCBI Sequence Read Archive (SRA) database under accession number PRJNA635750. Hi-C reads are

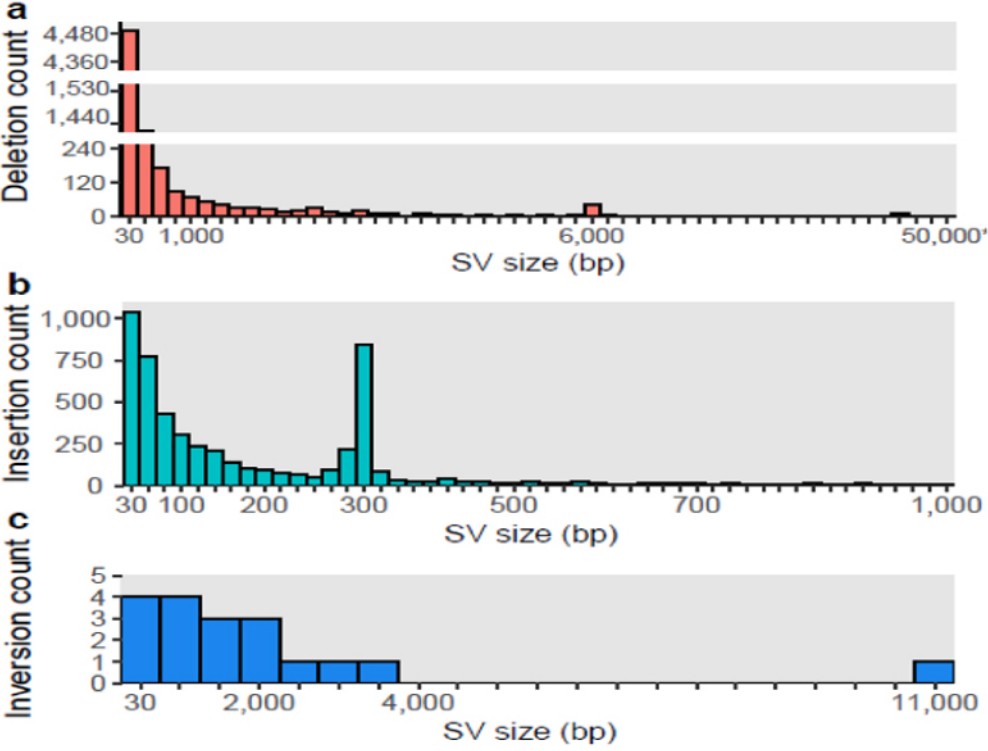

**Figure 6.** Complete size distribution of the consensus (a) deletions, (b) insertions, and (c) inversions found in LT1. Number of bins: (a) 54, (b) 50, (c) 21, respectively. Gaps on deletions *Y* axis are from 250 to 1,198 and from 1,550 to 4,000, aiming to reduce the overrepresentation of the first two bins.

deposited in Genbank with the SRA accession number SRR18003204 and BioProject number PRJNA635750. The LT1 assembly (including an mtDNA assembly), genome browser, BLAST database and variant calling data can be accessed via the LT1 webpage [40]. Additional data and processed results are also available from the *GigaScience* GigaDB repository [41].

## EDITOR'S NOTE

This study was part of the Korean Personal Genome Project (KPGP, also known as PGP-Korea) carried out by the Genome Research Foundation in Korea.

## DECLARATIONS
## LIST OF ABBREVIATIONS

BGI: Beijing Genomics Institute; bp: base pair; BUSCO: Benchmarking Universal Single-Copy Orthologs; CNV: copy number variation; DGV: Database of Genomic Variants; GB: gigabyte; Gbp: gigabase pair; GRCh38: Genome Reference Consortium Human Build 38; KOGIC: Korean Genomics Center; Mbp: megabase pair; mtDNA: mitochondrial DNA; NCBI: National Center for Biotechnology Information; ONT: Oxford Nanopore Technologies; QV: quality value; SNP: single nucleotide polymorphism; SRA: Sequence Read Archive; SV: structural variation; UNIST: Ulsan National Institute of Science and Technology; WGS: whole genome sequencing.

## ETHICAL APPROVAL

This study was approved by the Institutional Review Board of the Genome Research Foundation (reference: IRB-REC- 20101202 – 001). The anonymous sample donor provided informed consent to participate in whole genome sequencing and the following analysis in compliance with the Declaration of Helsinki. Informed consent was recorded by their signing of a written consent form.

## CONSENT FOR PUBLICATION

The consent form signed by the anonymous sample donor included a section about data publication, to which the sample donor specifically consented.

## COMPETING INTERESTS

DB is an employee of Geromics Inc. CK and JA are employees of Clinomics Inc., where JB is a founder and a CEO of Clinomics USA and Clinomics Inc. CK, JA, and JB all have an equity interest in the company. All other authors declare that they have no competing interests.

## FUNDING

This work was supported by the U-K BRAND Research Fund (grant number 1.200108.01) of Ulsan National Institute of Science & Technology (UNIST), and the research project was funded by the UNIST Ulsan City Research Fund (grant number 1.200047.01). This work was also supported by the Promotion of Innovative Businesses for Regulation-Free Special Zones funded by the Ministry of SMEs and Startups (MSS, Korea; grant numbers: P001193 (1425157301) (2.220036.01). This work was also supported by the Establishment of Demonstration Infrastructure for Regulation-Free Special Zones fund (MSS, Korea) (P0016191) (2.220037.01) and (1425157253) by the Ministry of SMEs and Startups. JB and CK were partially supported by Clinomics Inc. and the Genome Research Foundation.

## AUTHORS' CONTRIBUTIONS

JB and AB initiated the project as an openfree genome project. HK, AB, SJ, YK, CK, CY, and JA were in charge of methodology, formal analysis and visualization. AB and HK wrote the manuscript under the supervision of JB. JB was in charge of supervision, funding acquisition, and resources. JB, DB, AB, HK, SJ, CY, and JE contributed to the manuscript editing process and critical revisions. All authors read and approved the finalized manuscript.

## ACKNOWLEDGEMENTS

We thank GenomeLab, the Personal Genomics Institute of the Genome Research Foundation, and Korean Genomics Center (KOGIC) members for providing technical assistance and discussions. We also thank the Korea Institute of Science and Technology Information (KISTI) that provided us with the Korea Research Environment Open NETwork (KREONET).

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
