## [Reviewer Report]

Comments on revised manuscriptI am happy with the changes and I think the article is worth publishing in Gigabytes. However, I think one main point needs further clarity. Since this is mostly about a new dataset and assembly, the authors should make it very clear to the reader what they did. I think the title is still misleading in this respect. In it the authors refer to an "assembly with short and long reads combined with Hi-C data". This is not how one would generally refer to such an assembly in the community as it reads as if a short-read based assembly was complemented with long reads (gap filling?) and hic reads (phasing?). I suggest rephrasing as "a ONT long-read-based assembly scaffolded with Hi-C data and polished with short reads". The confusion/ambiguity about this is further reinforced in the text. I think the authors should make and extra effort reading the text to make sure the genome assembly terminology is consistent with the state of the art and therefore very clear to the reader. For instance, in the abstract the authors say that the assembly was constructed using 57x ultra-long nanopore reads. I think this is incorrect. Ultra-long nanopore reads are usually defined as reads >100kbp. I don't think the authors filtered their dataset for ultralong and this should be corrected. Indeed, it would be interesting to know what fraction of ultralong reads are available in their 57x dataset.

---

## [Reviewer Report]

Comments on revised manuscriptThe authors did not address any of my previous question, which is unacceptable.
In addition, http://lithuaniangenome.com genome website is not avaible to the reviewers. Moreover, it seems that the mitochondria was not correctly assembled (no mitochondria data was mentioned throughout the paper). Mitochondria miassembly is very common in the private genome assembly (Human Genetics volume 138, pages757–769 (2019)), which undervalues its applicability as reference genome.
In my opinion, this paper is not qualified enough for the journal of GigaByte.

---

## [Reviewer Report]

Upload additional filesDRR-202112-01/form/LT1_MS_GigaByte_20211123_HS_GF.docxReviewer name and names of any other individual's who aided in reviewer Giulio FormentiDo you understand and agree to our policy of having open and named reviews, and having your review included with the published papers. (If no, please inform the editor that you cannot review this manuscript.)YesIs the language of sufficient quality?YesPlease add additional comments on language quality to clarify if needed
A few minor typos to correct, highlighted in the revised manuscriptAre all data available and do they match the descriptions in the paper? YesAdditional CommentsAre the data and metadata consistent with relevant minimum information or reporting standards? See GigaDB checklists for examples <a href="http://gigadb.org/site/guide" target="_blank">http://gigadb.org/site/guide</a>YesAdditional CommentsIs the data acquisition clear, complete and methodologically sound?YesAdditional CommentsIs there sufficient detail in the methods and data-processing steps to allow reproduction?YesAdditional CommentsYes, but please revise as per my commentsIs there sufficient data validation and statistical analyses of data quality? YesAdditional CommentsIs the validation suitable for this type of data?YesAdditional CommentsIs there sufficient information for others to reuse this dataset or integrate it with other data?YesAdditional CommentsAny Additional Overall Comments to the AuthorRecommendationMinor Revision

---

## [Reviewer Report]

Reviewer name and names of any other individual's who aided in reviewer /Do you understand and agree to our policy of having open and named reviews, and having your review included with the published papers. (If no, please inform the editor that you cannot review this manuscript.)YesIs the language of sufficient quality?NoPlease add additional comments on language quality to clarify if needed
Are all data available and do they match the descriptions in the paper? NoAdditional CommentsHi-C data was not deposited.Are the data and metadata consistent with relevant minimum information or reporting standards? See GigaDB checklists for examples <a href="http://gigadb.org/site/guide" target="_blank">http://gigadb.org/site/guide</a>YesAdditional CommentsIs the data acquisition clear, complete and methodologically sound?NoAdditional CommentsThe quality of the nanopore sequencing datasets was not evaluated. The error correction using short-read sequencing was not clear. It seems not necessary to use Hi-C data for the assembly.Is there sufficient detail in the methods and data-processing steps to allow reproduction?NoAdditional Commentserror correction was not clear.Is there sufficient data validation and statistical analyses of data quality? NoAdditional CommentsIs the validation suitable for this type of data?YesAdditional CommentsNo validation of the variants was performed. The authors used multiple SNV detection algorithms and got quite different results. They should experimentally validate which one is better.Is there sufficient information for others to reuse this dataset or integrate it with other data?NoAdditional CommentsIt is difficult to reuse it. There's little annotation done.Any Additional Overall Comments to the AuthorI don't understand why the authors chose to sequence a woman. As a reference of a certain ethnic, complete chromosomes are needed, which means a man (XY) is necessary.RecommendationReject (Unsound or Unusuable)